# Electrical switching between exciton dissociation to exciton funneling in MoSe$_2$/WS$_2$ heterostructure

Yuze Meng[1,2,6], Tianmeng Wang[1,6], Chenhao Jin [3,6], Zhipeng Li [1], Shengnan Miao[1], Zhen Lian[1], Takashi Taniguchi[4], Kenji Watanabe [4], Fengqi Song [2✉] & Su-Fei Shi [1,5✉]

The heterostructure of monolayer transition metal dichalcogenides (TMDCs) provides a unique platform to manipulate exciton dynamics. The ultrafast carrier transfer across the van der Waals interface of the TMDC hetero-bilayer can efficiently separate electrons and holes in the intralayer excitons with a type II alignment, but it will funnel excitons into one layer with a type I alignment. In this work, we demonstrate the reversible switch from exciton dissociation to exciton funneling in a MoSe$_2$/WS$_2$ heterostructure, which manifests itself as the photoluminescence (PL) quenching to PL enhancement transition. This transition was realized through effectively controlling the quantum capacitance of both MoSe$_2$ and WS$_2$ layers with gating. PL excitation spectroscopy study unveils that PL enhancement arises from the blockage of the optically excited electron transfer from MoSe$_2$ to WS$_2$. Our work demonstrates electrical control of photoexcited carrier transfer across the van der Waals interface, the understanding of which promises applications in quantum optoelectronics.

[1] Department of Chemical and Biological Engineering, Rensselaer Polytechnic Institute, Troy, NY 12180, USA. [2] National Laboratory of Solid State Microstructures, Collaborative Innovation Center of Advanced Microstructures, and School of Physics, Nanjing University, 210093 Nanjing, P. R. China. [3] Kavli Institute at Cornell for Nanoscale Science, Cornell University, Ithaca, NY 14853, USA. [4] National Institute for Materials Science, 1-1 Namiki, Tsukuba 305-0044, Japan. [5] Department of Electrical Computer & Systems Engineering, Rensselaer Polytechnic Institute, Troy, NY 12180, USA. [6] These authors contributed equally: Yuze Meng, Tianmeng Wang, Chenhao Jin. ✉email: songfengqi@nju.edu.cn; shis2@rpi.edu

Two-dimensional (2D) semiconductors are promising candidates for light-harvesting and optoelectronic applications[1–5] due to their strong light–matter interaction from excitonic responses[6–13]. Their atomically thin nature further enables engineering exciton dynamics and energy relaxation pathways through ultrafast carrier transfer across 2D van der Waals (vdW) interfaces[14–21]. In particular, a vdW heterostructure can, respectively, dissociate electrons and holes into separate layers or funnel excitons to one layer with a type II or type I band alignment[15,21–28]. It is highly desirable to achieve both functions in a single device in an electrically reconfigurable way. However, to the best of our knowledge, this has not been demonstrated yet. Here we demonstrate reversible electrical switching between exciton dissociation and funneling in a $MoSe_2/WS_2$ heterostructure device. We show that the electron transfer from $MoSe_2$ to $WS_2$ can be blocked by efficient gating of the $LaF_3$ substrate, leading to a transition between photoluminescence (PL) quenching to PL enhancement for the $MoSe_2$ A exciton emission. The ability to electrically control interlayer charge transfer pathways ushers in application concepts, such as light switch and energy steering.

## Results

**Charge transfer in the $MoSe_2/WS_2$ heterostructure.** We construct the $MoSe_2/WS_2$ heterostructure on the $LaF_3$ substrate through a layer-by-layer dry transfer technique[29], and the heterostructure is also capped by a thin layer of hexagonal boron nitride (BN) on the top. A typical $MoSe_2/WS_2$ heterostructure on the $LaF_3$ substrate is shown in Fig. 1a. The overlapped region of the monolayer $MoSe_2$ and $WS_2$ forms the $MoSe_2/WS_2$ heterojunction. We use few-layer flakes of graphene to contact both the monolayer $MoSe_2$ and $WS_2$, and a schematic of the device is shown in Fig. 1b. The heterostructure can be gated through the $LaF_3$ substrate as the back gate, which provides efficient control of doping through the double layer[30], as schematically shown in Fig. 1b. Typical PL spectra for different regions of the device are shown in Fig. 1c, with the continuous wave (CW) laser excitation centered at 2.331 eV and a power of 100 μW. Without applying any gate voltage, the PL from the $MoSe_2/WS_2$ heterojunction (red line in Fig. 1c) exhibits quenching of both the PL at the $WS_2$ A exciton resonance (~1.979 eV) and $MoSe_2$ A exciton resonance (~1.548 eV), compared with that of the monolayer $WS_2$ (blue line

in Fig. 1c) and the monolayer $MoSe_2$ (black line in Fig. 1c), respectively (see Supplementary Note 4). This simultaneous quenching of PL at both $MoSe_2$ and $WS_2$ A excitons was observed in all the heterostructures we constructed, including three $MoSe_2/WS_2$ heterostructures on $SiO_2/Si$ substrate and seven heterostructures on $LaF_3$ in the absence of the gate voltage (see Supplementary Notes 1 and 7). It thus suggests a type II alignment for the as-prepared $MoSe_2/WS_2$ heterostructures, and the PL quenching is a result of the optically excited electron transferred to the $MoSe_2$ layer and hole transferred to the $WS_2$, according to the band alignment[31,32] shown schematically in the inset of Fig. 1c. It is interesting to note that the quenching of $MoSe_2$ A exciton PL is significantly less than that of the $WS_2$ A exciton in the heterojunction region. While the integrated PL of the $WS_2$ A exciton in the heterojunction is quenched by more than one order of magnitude smaller, the integrated PL intensity of $MoSe_2$ A exciton in the heterojunction is only slightly quenched, being ~70% of that from the monolayer $MoSe_2$ (Fig. 1c). The significantly less quenching of $MoSe_2$ A exciton PL can be understood from the relative band alignment shown in the inset of Fig. 1c. In the type II alignment configuration, the conduction band minimum (CBM) of the $WS_2$ is only slightly lower than that of the $MoSe_2$ according to the theoretical calculations[1,32]. The thermal equilibrium of the two CBMs at room temperature therefore allows a certain population of electrons in the CBM of the $MoSe_2$ even though the CBM of the $WS_2$ is the lower energy state for electrons in the heterojunction region. The small energy difference between the two CBMs offers the opportunity for us to apply an efficient electrostatic gating to manipulate the optically excited carrier transfer across the $MoSe_2/WS_2$ interface. We achieve that by using the $LaF_3$ as the ionic back gate, which has been proven to efficiently gate 2D materials though a double layer[30] (schematically shown in Fig. 1b).

**Gate-dependent PL enhancement in $MoSe_2/WS2$.** To reveal the effect of gate-controlled carrier transfer across the heterojunction, we then investigate the PL spectra around the $MoSe_2$ A exciton resonance as a function of the gate voltage for both monolayer $MoSe_2$ (Fig. 2a) and $MoSe_2/WS_2$ heterojunction (Fig. 2b) (see Supplementary Note 5). The CW laser excitation centered at 2.0 eV (620 nm) with a power of 100 μW was used to obtain the PL spectra shown in Fig. 2a, b. This excitation photon energy is

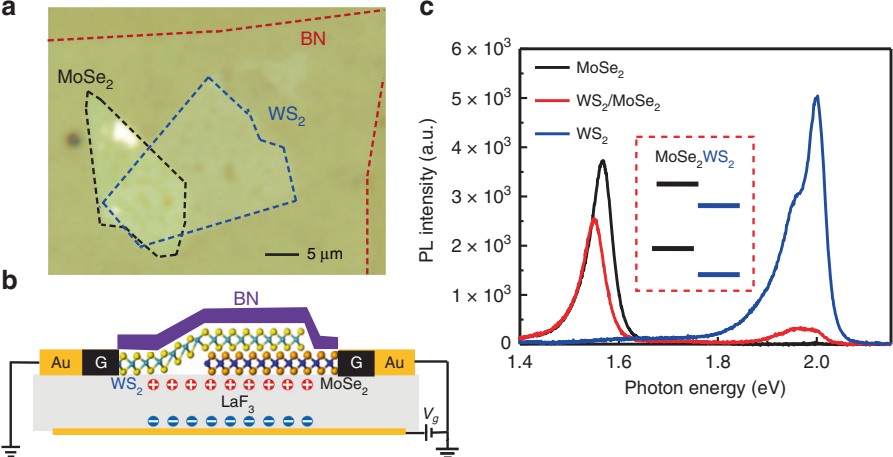

**Fig. 1 Monolayer $MoSe_2/WS_2$ heterostructure device. a** Optical microscopic image of the monolayer $MoSe_2/WS_2$ heterostructure, capped with a few-layer h-BN layer. Scale bar: 5 μm. **b** Schematic of the $MoSe_2/WS_2$ heterostructure device, contacted by few-layer graphene electrodes and gated by the ionic substrate $LaF_3$. **c** Typical room temperature PL spectra from regions of the monolayer $MoSe_2$ (black), monolayer $WS_2$ (blue), and $MoSe_2/WS_2$ heterojunction (red), with no gate voltage applied. Inset: schematic representation of the type II band alignment of the $MoSe_2/WS_2$ heterostructure.

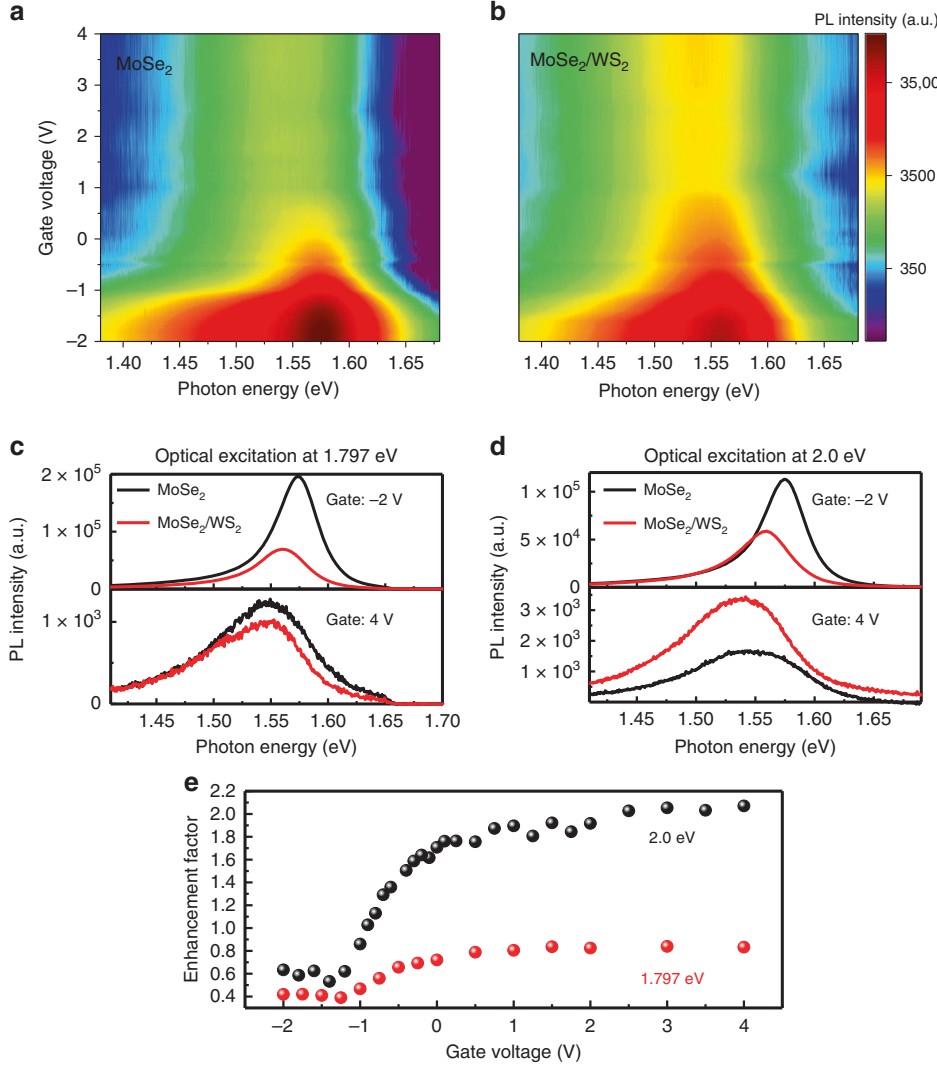

**Fig. 2 Gate voltage-tunable transition from PL quenching to PL enhancement in the MoSe₂/WS₂ heterojunction. a** The color plot of the PL spectra for monolayer MoSe₂ and **b** the color plot of the PL spectra for MoSe₂/WS₂ heterostructure region as a function of the gate voltage, under the continuous wave (CW) photoexcitation centered at 2.0 eV and with the excitation power of 100 μW. The color represents the integrated PL intensity at MoSe₂ A exciton resonance. All spectra were taken at room temperature. **c**, **d** are PL spectra of the monolayer MoSe₂ (black) and MoSe₂/WS₂ heterojunction (red) under the CW photoexcitation centered at 1.797 and 2.0 eV, respectively. The excitation power for both **c** and **d** is 100 μW. **e** The experimentally extracted PL enhancement factor as a function of the gate voltage for the photoexcitation centered at 2.0 eV (black dots) and 1.797 eV (red dots).

large enough to excite A excitons in both monolayer MoSe₂ (1.548 eV) and WS₂ (1.979 eV). The PL intensity from the WS₂ A exciton is drastically quenched in the heterojunction, and we focus on the PL intensity of the MoSe₂ A exciton for both the monolayer (Fig. 2a) and heterojunction region (Fig. 2b). We can see from the color plots (Fig. 2a, b) that, although the MoSe₂ A exciton PL intensity is weaker in MoSe₂/WS₂ heterojunction (Fig. 2b) than in the monolayer MoSe₂ (Fig. 2a) for the gate voltage from ~−2 to −1 V, the PL is stronger in the heterojunction than in the monolayer MoSe₂ at the gate voltage >0 V. This relative PL ratio from quenching to enhancement transition is clearly illustrated in the PL spectra in Fig. 2d, which combine the line cuts of Fig. 2a, b at the gate voltage −2 and 4 V. To better understand the PL behavior change, we define the PL enhancement factor (EF) as $EF = I_{Heter}/I_{MoSe_2}$, where $I_{Heter}$ ($I_{MoSe_2}$) is the integrated MoSe₂ A exciton PL intensity in the MoSe₂/WS₂ heterojunction (monolayer MoSe₂). EF as a function of the gate voltage for the photoexcitation centered at 2.0 eV is shown in

Fig. 2e (black dots), which shows that EF is almost a constant between the gate voltage of −2 to −1 V (EF ~ 0.6) but quickly rises to ~1.8 at the gate voltage 0 V, and it remains largely a constant as the gate voltage is further increased.

It is interesting to note that this observation is sensitive to the excitation photon energy, and the results are distinctively different for the CW photoexcitation of the same power (100 μW) but centered at 1.797 eV (690 nm), which is below the A exciton resonance energy of WS₂ but above that of MoSe₂. At the gate voltage of −4 V, we observe PL quenching at MoSe₂ A exciton resonance in the heterojunction (Fig. 2d), similar to the scenario with the photoexcitation at 2 eV (Fig. 2c). However, as we increase the gate voltage to 4 V, we do not observe the PL enhancement of the MoSe₂ A exciton in the heterojunction, even though the PL intensity is quite close to that of the monolayer MoSe₂ (Fig. 2d). A detailed gate-dependence study of the photoexcitation centered at 1.797 eV also results in quantitative EF as shown in Fig. 2e (red dots), which shows a similar step function behavior as the case of

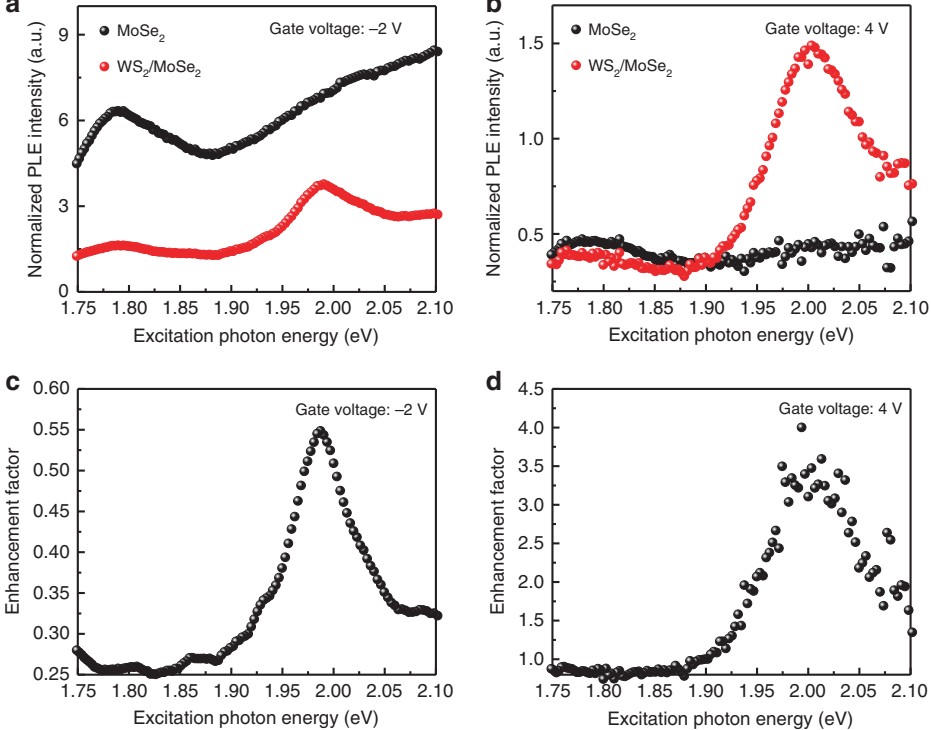

**Fig. 3 PLE spectra of the monolayer MoSe₂ and MoSe₂/WS₂ heterostructure for different gate voltages. a, b** are integrated PL intensity at MoSe₂ A exciton resonance as a function of the excitation photon energy for monolayer MoSe₂ (black) and MoSe₂/WS₂ heterojunction (red) regions at the gate voltage of −2 V (**a**) and the gate voltage of 4 V (**b**), respectively. **c, d** are PL enhanced factor for the gate voltage of −2 V (**c**) and 4 V (**d**), respectively.

photoexcitation centered at 2.0 eV, but the maximum value of EF is smaller and never exceeds 1.

**PL excitation (PLE) spectroscopy of EF in MoSe₂/WS₂.** Since the observed PL EF at MoSe₂ A exciton resonance is sensitive to the excitation photon energy, we then perform a detailed PLE spectroscopic study. The integrated PL intensity at the MoSe₂ A exciton resonance for monolayer MoSe₂ (black) and MoSe₂/WS₂ heterojunction (red) are plotted as a function of the excitation photon energy in Fig. 3a, b for the gate voltage of −2 and 4 V, respectively. Figure 3c shows the EF for the gate voltage of −2 V, and when both MoSe₂ and WS₂ are intrinsic, the MoSe₂ A exciton PL intensity is always quenched for different excitation photon energies, with the peak value of 0.55 at the excitation photon energy of ~2.0 eV. However, at the gate voltage of 4 V, the EF is largely flat and slightly <1 (~0.8) when the excitation photon energy is <1.9 eV. When the excitation photon energy >1.9 eV, the PL enhancement effect starts to occur with the EF >1. The EF reaches the maximum value when the excitation photon energy is about 2.0 eV. The excitation photon energy for the peaked EF value in Fig. 3c, d coincides with the A exciton resonance of monolayer WS₂, which suggests that photoexcited carrier transfer from WS₂ to MoSe₂ plays the central role in the PL enhancement in the MoSe₂/WS₂ heterojunction, which, as a result, explains the lack of the PL enhancement with the off-resonance excitation at 1.797 eV (Fig. 2d).

It is worth noting that, even with the CW photoexcitation centered at 2.0 eV, we have not observed that the PL EF exceeds 1 for three MoSe₂/WS₂ heterojunction devices fabricated on SiO₂/Si substrate (300-nm-thick thermal oxide), with the silicon back gate voltage as high as 80 V (see Supplementary Note 1). This observation suggests that the efficient gating from LaF₃ is essential for realizing the PL enhancement in the heterojunction.

Previous work has shown that the LaF₃ back gate should be at least >100 times more efficient than the silicon back gate with 300 nm thermal oxide[30].

## Discussion

The experimental observation can be understood theoretically by considering the gate-dependent carrier distribution in the heterostructure. Taking into account the quantum capacitance of the monolayer MoSe₂ and WS₂, for the device configuration shown in Fig. 1b, the effective capacitance model can be schematically shown as the inset of Fig. 4 (MoSe₂ being the bottom layer, and detailed derivation in Supplementary Note 3). Here $C_{Q1}(C_{Q2})$ are the quantum capacitance of monolayer MoSe₂ (WS₂), $C_{G1}$ is the geometry capacitance between MoSe₂ and the LaF₃ back gate, and $C_{G2}$ is the geometry capacitance between MoSe₂ and WS₂. For qualitative understanding, we consider zero-temperature case here (see Supplementary Note 2 for the discussion of the finite temperature case, which does not qualitatively change the picture). Owing to the large energy difference between the VBMs of MoSe₂ and WS₂, the hole transfer from WS₂ to MoSe₂ (when WS₂ is optically excited) is always ~100%. As a result, we focus on the gate dependence of the electron transfer from MoSe₂ to WS₂. As shown in Fig. 4, when the gate voltage is at point A (e.g., −2 V for the device 2 shown in Fig. 2), both the MoSe₂ and WS₂ layers are intrinsic and with the quantum capacitance of zero. As a result, the gate voltage is dropped only on the quantum capacitance and the band alignment is determined by the work function of each layer. The type II alignment (shown at point A in Fig. 4) determines that the optically excited electron in MoSe₂ will transfer to WS₂, reducing the electron density in the MoSe₂ layer in the heterojunction, compared to the case of the bare monolayer MoSe₂. In addition, with (on-resonance excitation) and without (off-resonance excitation) the hole transfer from WS₂ to MoSe₂,

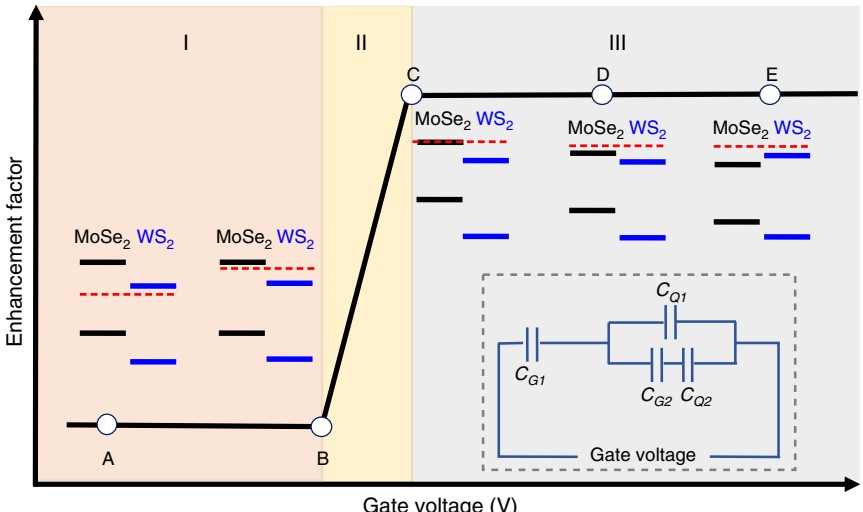

**Fig. 4 Theoretical understanding of the PL quenching to PL enhancement transition.** The enhancement factor as a function of the gate voltage clearly exhibits three distinct regions. Schematics of band alignment of the MoSe$_2$/WS$_2$ heterojunction, along with the Fermi energy level (dashed line), are labeled for different points to explain the different PL enhancement factor behaviors. Inset: schematic of the effective capacitance circuit of the MoSe$_2$/WS$_2$ heterojunction with the device configuration shown in Fig. 1b.

the electron density in the MoSe$_2$ layer in the heterojunction is always less than the hole density. As a result, the electron is the minor carrier that determines the available MoSe$_2$ A exciton density. The reduced electron density thus leads to the quenching of MoSe$_2$ A exciton PL in the heterojunction.

When the WS$_2$ layer starts to get electron-doped (point B), the number of optically excited electrons transferred from the MoSe$_2$ to WS$_2$ in the heterojunction region will be modulated by the gate voltage. For simplicity, we can use the off-resonance excitation as an example. The charge transfer from MoSe$_2$ to WS$_2$, $\Delta Q$, can be obtained from the following equation according to the effective capacitance model[33] (inset of Fig. 4):

$$\frac{Q - \Delta Q}{C_{Q1}} = \frac{\Delta Q}{C_{Q2}} + \frac{\Delta Q}{C_{G2}}, \quad (1)$$

where $Q$ is the total charge of optically excited electrons in the MoSe$_2$ layer of the MoSe$_2$/WS$_2$ heterojunction. Reorganization of Eq. (1) results in the expression of $\Delta Q$ as:

$$\frac{\Delta Q}{Q} = \frac{1}{1 + \frac{C_{Q1}}{C_{Q2}} + \frac{C_{Q1}}{C_{G2}}}. \quad (2)$$

For gate voltage smaller than that of point B, $C_{Q1} = 0$ and hence $\Delta Q = Q$, which indicates that ~100% of the optically excited electron in MoSe$_2$ layer of the heterojunction region is transferred to WS$_2$. As a result, PL quenching of the MoSe$_2$ layer in the heterojunction is similar to that of A point (similar EF). As we move forward from point B, however, electron transfer will be less efficient due to the finite $C_{Q1}$ (i.e., finite density of states (DOS) at Fermi level in MoSe$_2$) and the PL quenching will be less significant. As the gate voltage is increased to point C, the doping further increases and the Fermi level is aligned with the conduction band of MoSe$_2$. Assuming a similar effective electron mass $m$ in WS$_2$ and MoSe$_2$, we have $C_{Q1} = C_{Q2} = C_Q$, where $C_Q = \frac{m}{\pi\hbar^2}$ is the DOS in 2D. Since $C_Q \gg C_{G2}$ (see Supplementary Note 2), from Eq. (2), we found that $\Delta Q \sim 0$ and optically excited electron transfer from MoSe$_2$ to WS$_2$ is blocked. The EF of MoSe$_2$ A exciton PL will therefore again be largely a constant, with the value of 1 (Fig. 4) for the off-resonance excitation in the ideal scenario.

The electron transfer in the on-resonance scenario can be understood in a similar fashion (see Supplementary Note 2), with similar PL quenching (EF < 1) from point A to B. However, when MoSe$_2$ is sufficiently doped (point C), optically induced holes in the MoSe$_2$ layer become the minor carrier that determines the MoSe$_2$ A exciton density. For the on-resonance excitation, the WS$_2$ layer is also excited and we have nearly 100% of the optically excited holes transfer from WS$_2$ to MoSe$_2$. Therefore, the A exciton density in the MoSe$_2$ layer in the heterojunction is enhanced, giving rise to the PL enhancement with a largely constant EF > 1. We thus conclude that, for both the off-resonance and on-resonance excitation, the qualitative gate dependence of EF will be of the form shown in Fig. 4. Particularly, EF will show an abrupt increase around specific gate voltage (region II) and remain largely constant on either side. On the low voltage side (region I), EF should be <1; and on the high gate voltage side (region III), EF = 1 for the off-resonance excitation and EF > 1 for the on-resonance excitation.

The theoretical prediction in Fig. 4 is in excellent agreement with our experimental observation in Fig. 2e. The experimentally observed EF as a function of the gate voltage can be clearly divided into three regions, similar to a step function for both the on-resonance and off-resonance excitation as predicted by Fig. 4. The EF for the on-resonance excitation (photoexcitation at 2.0 eV) in region III shows an EF ~ 2.0, while the EF for the off-resonance excitation (photoexcitation at 1.797 eV) in the region III is about 0.8. The EF of <1 for the off-resonance case is probably due to decreased quantum efficiency in the heterojunction from the different dielectric environment.

The consideration of the finite temperature effects is included in Supplementary Note 2, and it gives qualitatively similar results as in Fig. 4. Interestingly, we found that, for large enough gate voltage, the charge accumulated on the $C_{G2}$ will give rise to a large energy shift between the MoSe$_2$ and WS$_2$, which switches the type II alignment to a type I alignment configuration, as shown schematically by the inset at point E in Fig. 4. The efficient ionic gating thus not only allows the control of optically excited carrier transfer across the atomically sharp interface but also leads to the possibility of modifying the alignment type reversibly. The associated fundamental understanding will enable quantum

optoelectronics based on transition metal dichalcogenide (TMDC) vdW heterostructures.

## Method

**Device fabrication.** The MoSe$_2$/WS$_2$ heterostructure devices were fabricated through a layer-by-layer dry transfer technique[29]. More specifically, each of the monolayer TMDC was sequentially transferred to the LaF$_3$ substrate, and a final BN flake was used to cap the heterostructure. Two pieces of few-layer graphene were used as the electrodes to contact the monolayer MoSe$_2$ and WS$_2$ layer separately, and both were grounded during the measurements, as schematically shown in Fig. 1b. The final devices were annealed in vacuum at 100 °C for 3 h.

**Optical measurements.** All the optical measurements in this work were performed at room temperature. The micro-PL measurements were performed with a home-built confocal microscope, in which the excitation lasers were focused to a spot size of ~2 μm. The PLE spectra were taken with a supercontinuum white laser (Fianium), and the filtered light (with bandwidth ~4 nm) was used as the excitation source.

## Data availability

The data that support the findings of this study are available from the authors on reasonable request, see "Author contributions" for specific data sets.

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

## Acknowledgements

T.W. and S.-F.S. acknowledge support from ACS PRF through grant 59957-DNI10. S.M., Z.Li, and S.-F.S. acknowledge support from AFOSR through Grant FA9550-18-1-0312. Z.Lian and S.-F.S. acknowledge support from NYSTAR through Focus Center-NY–RPI Contract C150117. The device fabrication was supported by the Micro and Nanofabrication Clean Room (MNCR) at Rensselaer Polytechnic Institute (RPI). K.W. and T.T. acknowledge support from the Elemental Strategy Initiative conducted by the MEXT, Japan and the CREST (JPMJCR15F3), JST. F.S. acknowledges support from the National Natural Science Foundation of China (No. U1732273). S.-F.S. also acknowledges the support from a KIP grant from RPI and a VSP grant from NHMFL.

## Author contributions

S.-F.S. conceived the experiment. Y.M. and Z.Lian fabricated the devices. Y.M. and T.W. performed the measurements. S.-F.S., T.W., Y.M., C.J., S.M., Z.Li and F.S. analyzed the data. T.T. and K.W. grew the BN crystals. S.-F.S. supervised the project. S.-F.S., T.W., C.J., S.M., and Y.M. wrote the manuscript with input from all the other co-authors. All authors discussed the results and contributed to the manuscript.

## Competing interests

The authors declare no competing interests.
