## [Peer Review File · Nature Communications]

Reviewers' comments:

Reviewer #1 (Remarks to the Author):

In this manuscript, Meng and coauthors reported a transition from PL quenching to PL enhancement in MoSe₂/WS₂ heterostructure (compared to MoSe₂ monolayer) by electric gating and explained with a quantum capacitance model. The result is certainly interesting and worth further exploration. The manuscript is well written. I would recommend the publication of this manuscript after the following concerns have been addresses.

1) it's interesting that the PL quenching of MoSe₂ is so small in heterostructure. The author ascribed it to small CBM offset. A similar redshifted and strong PL has also been observed in this heterostructure, which has been assigned to CT emission (Nanoscale, 2015, 7, 17523–17528). If CBM offset is very small, intra- and inter- exciton emission could be very close in energy. Could author justify the nature this emission or comment on this. TRPL measurement should help. Also is this quenching similarly small in different samples? It would be more informative to show some statistics. Another concern regarding this small quenching is whether this is just due to the change of doping level but not band alignment? e.g. more exciton in MoSe₂ and more trion in heterostructure.

2) In this manuscript, the author compares the PL intensity of MoSe₂ in monolayer and in heterostructure to define the absolute enhancement factor (e.g. > 1 for enhancement, or <1 for quenching). Usually, there could be large heterogeneity between different spatial locations for this kind of stacked sample such that this definition of enhancement factor could be not that reliable. For example, the dielectric environment of MoSe₂ in heterostructure is different from that in monolayer. So the PL properties including the energy and intensity of MoSe₂ should have changed and '1' might not be a real board line. Another tricky thing about this enhancement factor is when applying gating, the PL intensity of MoSe₂ monolayer is also changing. If MoSe₂ monolayer and heterostructure have different doping level, this calculation of enhancement factor at same gating voltage might not as good as enhancement factor calculated at same MoSe₂ doping density. Considering small PL intensity change in MoSe₂, this enhancement factor might also have some uncertainty. The author may want to show results from multiple samples.

3) Another way to check energy transfer contribution on PL is relative enhancement (relative to MoSe₂ itself in heterostructure, not in monolayer). This can be seen in Figure 3 c and 3d. Compared with below WS₂ bandgap excitation (e.g. 1.8 eV), exciting WS₂ (e.g. 2 eV) leads to MoSe₂ PL enhancement under either -2 V or 4 V gating. To me, there indicates energy funnel from WS₂ to MoSe₂ regardless of gating voltage. The author might explain or comment on this.

4) According to Fig. 4, PL enhancement occurs in n-doped region where fermi level is above CB. This should show some optical signature on PL and absorption spectra. The authors might should find some signature to further support proposed picture.

Reviewer #2 (Remarks to the Author):

In the manuscript titled "Electrical switching between exciton dissociation to exciton funnelling in MoSe₂/WS₂ heterostructure", Meng et al. report on the gate dependence of the photoluminescence (PL) and photoluminescence excitation (PLE) of a gated MoSe₂/WS₂ heterostructure. Van der Waals heterostructures composed by monolayer transition metal dichalcogenides exhibit a type II band alignment, wherein the maximum of the valence band resides in one material, while the minimum of the conduction band is in the other layer. This leads to the rapid formation of an excitonic complex known as interlayer exciton, created following efficient charge transfer upon photoexcitation. In the submitted manuscript, the authors tune the

intensity of the emission of the lowest lying PL peak, attributed to the recombination of the intralayer exciton of MoSe₂, by varying the gate voltage of the device. The key point to reach a doping level sufficient for the observation of this effect is the use of LaF₃ as the substrate. This ionic solid has been proven to provide very efficient gating, while maintaining a good stability. They propose a model, which relies on the quantum capacitance of the doped monolayers, to explain the experimental observations.

The results are novel and interesting. They have the potential to provide a significant impact to the field and, more generally, could be appreciated also by the broad readership of Nature Communications. The sample fabrication, experiments and data interpretation are conducted in a clear and consistent way. I would suggest to the authors to submit a revised version of the manuscript with some improvements and clarifications listed below to improve the quality of their work.

1. The first question which "spontaneously" arises when reading the paper is whether the authors gave a fair share to the hypothesis that the low energy peak in their PL spectrum of the heterostructure is related to the recombination of the interlayer exciton. This quasiparticle complex has already been reported in the heterostructure studied by the authors in Nano Letters 17, 5342 and ACS Nano 9, 6459 among others. Because of the peculiar band alignment of the two constituents, the PL energy of the interlayer exciton in this heterostructure is only slightly lower in energy than the resonance of the intralayer MoSe₂ exciton, and the energy shift between the interlayer exciton and the MoSe₂ exciton reported in the literature is compatible with what is shown in figure 1c of the submitted manuscript. Additionally, the enhanced intensity of the low energy PL peak described in the manuscript could be explained in terms of a stronger absorption when the excitation is performed in resonance with intralayer excitons. The authors do not mention at all the interlayer exciton as the origin of the low energy peak in the PL spectrum. I find this quite surprising and I suggest instead that a complete discussion is presented, with convincing arguments which rule out the role of the interlayer excitons. A possibility could be for example to show temperature dependent data, similar to what is shown in Nature Physics 14, 801. This measurement could help clarify the origin of the peaks in the PL spectrum.
2. Related to this, from lines 77-78, I seem to understand that all the measurements are performed at room temperature, although the authors never specify this explicitly. As per Nature journals template, they should add a "Method" section, where they indicate the experimental conditions they used and add more details concerning the device fabrication.
3. Always related to my first question: did the authors measure PLE vs gate voltage also for excitation energies >2eV? Does the intensity of the low energy peak stay constant or does it decrease?
4. It would be very instructive if the authors could quantify the doping level reached by using the LaF₃ substrate and compare them with SiO₂-based devices.
5. I was wondering what role can play the charged exciton in the absorption of WS₂. If the gating is so efficient, at a sufficient gate voltage there should be a "transfer" of oscillator strength from the neutral to the charged exciton, which could be for example visualized by performing gate dependent reflectivity contrast map (as in PRB 96, 085302, for example), which should in turn influence the energy at which the resonance is detected in PLE.
6. Given that there are no space restrictions, I think that the readability of the manuscript would be improved if the authors provided more detail about the derivation of the equivalent circuit shown as the inset of figure 4.
7. The authors wrote they fabricated seven devices with LaF₃ as the substrate. Did they perform similar measurements as a check on any other device? Do they reproduce their observations?
8. Small points: line 91. The authors write WSe₂, I suppose it should be WS₂. Lines 95-96. The authors write about an "evident" enhancement of the signal of MoSe₂. I think it is misleading for the reader: one might think that the enhancement occurs for the signal from the heterostructure when the voltage is changed, which is not the case. The authors refer to the larger intensity of the signal from the heterostructure as compared to the signal from MoSe₂ at the same gate voltage. I suggest to rephrase to avoid any confusion.

Reviewer #1 (Remarks to the Author):

In this manuscript, Meng and coauthors reported a transition from PL quenching to PL enhancement in MoSe₂/WS₂ heterostructure (compared to MoSe₂ monolayer) by electric gating and explained with a quantum capacitance model. The result is certainly interesting and worth further exploration. The manuscript is well written. I would recommend the publication of this manuscript after the following concerns have been addresses.

We sincerely thank the reviewer for the recognition of our work, and we provide a point-to-point response to the reviewer's comments as follows.

1) it's interesting that the PL quenching of MoSe₂ is so small in heterostructure. The author ascribed it to small CBM offset. A similar redshifted and strong PL has also been observed in this heterostructure, which has been assigned to CT emission (Nanoscale, 2015, 7, 17523–17528). If CBM offset is very small, intra- and inter- exciton emission could be very close in energy. Could author justify the nature this emission or comment on this. TRPL measurement should help. Also is this quenching similarly small in different samples? It would be more informative to show some statistics. Another concern regarding this small quenching is whether this is just due to the change of doping level but not band alignment? e.g. more exciton in MoSe₂ and more trion in heterostructure.

We thank the reviewer for the insightful question of the nature of PL from the heterojunction, whether it is from the intralayer exciton in MoSe₂ or interlayer exciton in MoSe₂/WS₂. We take the opportunity to elaborate here.

The fabrication of the hetero-bilayer of MoSe₂/WS₂ was through random stacking, which means that the K (K') valleys of the MoSe₂ and WS₂ are not aligned. This makes the charge transfer excitons (interlayer excitons) indirect bandgap in nature with momentum-mismatched electrons and holes. As a result, their radiative recombination rate and PL intensity should be strongly reduced.

We have performed the TRPL measurements at the MoSe₂ A exciton resonance for the monolayer MoSe₂ and MoSe₂/WS₂ heterojunction, as shown in Fig. R1. It is evident that the PL lifetime from the MoSe₂ monolayer and the MoSe₂/WS₂ hetero-bilayer are almost the same (the one from the heterojunction is actually slightly shorter), contradicting the expected longer lifetime from the interlayer exciton due to spatial separation of electrons and holes. In addition, the fitting shows a fast decay component of ~ 5 ps (Fig. R1), consistent with the previously reported lifetime of intralayer exciton but much shorter than the reported interlayer exciton lifetime in MoSe₂/WS₂ heterojunction (~ 80 ps) extracted from the transient absorption measurement (Nanoscale, 2015, 7, 17523–17528).

Figure R1. The time-resolved PL of the monolayer MoSe_2 and $\text{MoSe}_2/\text{WS}_2$ junction at room temperature. (a) TRPL spectra from the heterojunction (red dots), monolayer MoSe_2 (blue dots), and response from the laser (black). (b) and (c) and convolution fittings to the experimental TRPL using the laser response as the kernel for the heterojunction and monolayer MoSe_2 , respectively.

We have fabricated seven heterostructure devices on LaF_3 and three devices on SiO_2/Si in total. The data from devices on the LaF_3 substrate are shown in Fig. R2, and all the heterojunction showed PL quenching with no gate voltage applied, and the enhancement factor (EF) varying from 0.5 to 0.9. This systematic quenching behavior is different from the PL enhancement observed in the mentioned reference (Nanoscale, 2015, 7, 17523–17528), in which the PL enhancement was observed ($\text{EF} > 1$) as the signature of the CT exciton (interlayer exciton).

Figure R2. PL EF for different $\text{MoSe}_2/\text{WS}_2$ heterojunctions on the LaF_3 substrate.

Based on the above discussion, we confidently assign the lower energy PL peak in the heterojunction to the intralayer exciton of MoSe_2 , and we have added the related discussion to SI Supplementary Note 4. We thank the reviewer for helping us to improve our manuscript.

We also thank the reviewer for bringing up the concern of the doping effect on the quenching. We emphasize that the PL quenching to PL enhancement transition in this work was shown in a systematic study of gate dependence and excitation photon energy dependence, instead of relying on the reported value from one gate voltage. This systematic study rules out the possibility of the doping effect. For example, due to the efficient gating by the LaF₃ substrate, at the gate voltage of -2 V, both MoSe₂ and WS₂ should be intrinsic, evidenced by the strong PL (Fig. 2c,d of the main text). In this scenario, the MoSe₂ in the heterojunction is also charge-neutral. The quenching we observed at the gate voltage -2 V, therefore, cannot be ascribed to the doping difference. In addition, if the PL quenching in the heterojunction (MoSe₂ A exciton) compared with monolayer MoSe₂ is due to the doping difference, say MoSe₂ in the heterojunction is n-doped and bare monolayer MoSe₂ is intrinsic, we can always find a gate voltage to bring the MoSe₂ in the heterojunction to intrinsic by introducing holes. This will p-dope the bare MoSe₂ or at least bring the bare monolayer MoSe₂ to be intrinsic (the TMDCs studied here are difficult to be p-doped). As a result, we should see that, as we decrease the gate voltage (more p-doped), the PL from the hetero-junction should be at least comparable to that from the monolayer MoSe₂, or in other words, the EF will rise to a value close to 1 as we decrease the gate voltage. This contradicts our observation, which shows that the EF decreases monotonically as we reduce the gate voltage.

2) In this manuscript, the author compares the PL intensity of MoSe₂ in monolayer and in heterostructure to define the absolute enhancement factor (e.g. > 1 for enhancement, or <1 for quenching). Usually, there could be large heterogeneity between different spatial locations for this kind of stacked sample such that this definition of enhancement factor could be not that reliable. For example, the dielectric environment of MoSe₂ in heterostructure is different from that in monolayer. So the PL properties including the energy and intensity of MoSe₂ should have changed and '1' might not be a real board line. Another tricky thing about this enhancement factor is when applying gating, the PL intensity of MoSe₂ monolayer is also changing. If MoSe₂ monolayer and heterostructure have different doping level, this calculation of enhancement factor at same gating voltage might not as good as enhancement factor calculated at same MoSe₂ doping density. Considering small PL intensity change in MoSe₂, this enhancement factor might also have some uncertainty. The author may want to show results from multiple samples.

The reviewer is correct in noting that the borderline might not be 1, and simply compare the PL intensity itself at a fixed gate with certain photoexcitation might not reveal the real enhancement or quenching of the PL. Therefore, it is crucial to investigate the gate voltage dependence and photoexcitation energy dependence. As shown in Fig. 2d in the main text, the EF approach a constant at ~0.8 for the off-resonance excitation at 1.797 eV, for a broad range of gate voltage that corresponds to a highly n-doped region (gate voltage > 1 V). According to our theoretical understanding, this is the region that the PL from the heterojunction and monolayer MoSe₂ PL should be the same. Hence, the 0.8 is the borderline for PL enhancement and quenching for this particular case. The deviation from 1 could be simply due to the decreased quantum efficiency of the monolayer MoSe₂ in the heterojunction, which has a different dielectric environment. The

borderline also has some experimental uncertainty. From Fig. 3d of the main text, the borderline varies from 0.8 to 0.9 for different photoexcitation energy below the WS₂ A exciton resonance. However, our quenching to enhancement transition typically has an EF much smaller or larger than the borderline.

In our reply to comment 1 of reviewer 1, we have ruled out the possible contribution to quenching due to doping difference, and we have also shown the PLE data from another device in Fig. R10, in response to comment 7 by reviewer 2. The results are similar to our data shown in Fig. 3 in the main text.

We agree with the reviewer on the possible spatial inhomogeneity of stacked heterostructure devices. Therefore, we performed spatially resolved PL mapping for the heterojunction and monolayer MoSe₂ at different experimental conditions, shown in Fig. R3. It is evident that at the gate voltage of -3 V (Fig. R3a,b), when both MoSe₂ and WS₂ are intrinsic, the PL (MoSe₂ A exciton) from the heterojunction is also weaker than that from the monolayer MoSe₂, no matter whether the optical excitation can excite the WS₂ (2.33 eV) or not (1.80 eV). However, at the gate voltage of 4 V (Fig. R3c,d), when both MoSe₂ and WS₂ are highly n-doped, the PL from the heterojunction is stronger than the monolayer MoSe₂ when the optical excitation excites WS₂ (2.33 eV). It is clear that there is spatial inhomogeneity, which will eventually give rise to the uncertainty in the enhancement factor. However, the overall trend of enhancement is evident.

It is worth noting that the PL from the heterojunction appears to be weaker than the monolayer MoSe₂ when the optical excitation is below the A exciton resonance of WS₂ (1.80 eV), while theoretically, it should be the same. The reason is, as the reviewer describes, the borderline is less than 1, typically varying from 0.8 to 0.9. There are also a few spots that PL remain relatively strong (red spots in Fig. R3d), which we attribute to the poor coupling to the LaF₃ substrate, and the gate voltage was not effectively applied to the sample at those spots. We have included the spatial PL mapping in the revised SI as Supplementary Note 7.

Figure R3. Spatial PL mapping. (a) and (b) are the spatially resolved PL mapping at the gate voltage of -3 V with the photoexcitation centered at 2.33 eV and 1.80 eV, respectively. (c) and (d) are the spatially resolved PL mapping at the gate voltage of 4 V with the photoexcitation centered at 2.33 eV and 1.80 eV, respectively

3) Another way to check energy transfer contribution on PL is relative enhancement (relative to MoSe₂ itself in heterostructure, not in monolayer). This can be seen in Figure 3 c and 3d. Compared with below WS₂ bandgap excitation (e.g. 1.8 eV), exciting WS₂ (e.g. 2 eV) leads to MoSe₂ PL enhancement under either -2 V or 4 V gating. To me, there indicates energy funnel from WS₂ to MoSe₂ regardless of gating voltage. The author might explain or comment on this.

We thank the reviewer for the suggestion to check the energy transfer contribution. We have checked the data in Fig. 3a, b of the main text, and the reviewer is correct in that, compared with the excitation at 1.8 eV, the excitation at 2 eV leads to MoSe₂ PL enhancement for both gate

voltage of -2 V and 4 V. However, such comparison is complicated and cannot directly provide information on the charge/energy transfer pathways.

First, the absorption of MoSe₂ at 1.8 eV and 2 eV is not necessarily the same. The PL intensity difference can therefore be a consequence of different optical absorption.

In addition, if assuming the same absorption, a stronger PL with WS₂ excitation would be a natural consequence of carrier transfer regardless of the band alignment. If the MoSe₂/WS₂ bilayer forms a type I alignment (Fig. R4c), the excitation of WS₂ will obviously enhance (compare to monolayer MoSe₂) the PL of MoSe₂ as both electrons and holes in WS₂ will transfer to MoSe₂. If the MoSe₂/WS₂ bilayer forms a well-defined type II heterostructure with a large valence band offset (regardless of the doping level, Fig. R4a or Fig. R4b), exciting at WS₂ resonance would lead to two consequences: (1) The excited holes in WS₂ will transfer to MoSe₂; (2) Fewer electrons in MoSe₂ will transfer to WS₂ due to the increased electron density in WS₂ through optical excitation. As a result, both the electron and hole density in MoSe₂ will be higher when exciting WS₂ as compared to exciting MoSe₂ only, and the MoSe₂ PL intensity will be stronger.

Given the above complications, we have not used the comparison between PL under different excitations as an indication of the charge transfer pathway.

Figure R4. Schematics for type II alignment with intrinsic doping (a), type II alignment with high n-doping (b), and type I alignment with high n-doping (c).

4) According to Fig. 4, PL enhancement occurs in n-doped region where fermi level is above CB. This should show some optical signature on PL and absorption spectra. The authors might should find some signature to further support proposed picture.

We include here the PL spectra for the heterojunction and monolayer in Fig. R5. It is evident that the PL intensity decreases significantly as the gate voltage increases from -1 V to 4 V. This suggests strong n-doping of the MoSe₂, both in the heterojunction and monolayer. In addition, we observe a 20 meV redshift of the PL peak, which is consistent with the doping behavior of PL in MoS₂ at room temperature [Nature Materials 12, 207–211(2013)]. Note that due to the broad PL linewidth at room temperature, the exciton and trion peaks are not well-separated, and the emergence of trion manifests as a redshift in the PL peak [Nature Communications 4, 1474 (2013)].

In our next reply to comment 4 by reviewer 2, we also show the transport data on monolayer MoS₂ gated by LaF₃. It is shown that the monolayer MoS₂ on LaF₃ is strongly n-doped as the voltage is increased. In addition, the monolayer MoS₂ FET device on LaF₃ exhibits an insulator to metal transition at a relatively low gate voltage, confirming that the Fermi energy should be in the conduction band for the gate voltage we show (typically 4 V).

Figure R5. PL spectra for the heterojunction (a) and monolayer MoSe₂ (b) for different gate voltages.

Reviewer #2 (Remarks to the Author):

In the manuscript titled “Electrical switching between exciton dissociation to exciton funnelling in MoSe₂/WS₂ heterostructure”, Meng et al. report on the gate dependence of the photoluminescence (PL) and photoluminescence excitation (PLE) of a gated MoSe₂/WS₂ heterostructure. Van der Waals heterostructures composed by monolayer transition metal dichalcogenides exhibit a type II band alignment, wherein the maximum of the valence band resides in one material, while the minimum of the conduction band is in the other layer. This leads to the rapid formation of an excitonic complex known as interlayer exciton, created following efficient charge transfer upon photoexcitation. In the submitted manuscript, the authors tune the intensity of the emission of the lowest lying PL peak, attributed to the recombination of the intralayer exciton of MoSe₂, by varying the gate voltage of the device. The key point to reach a doping level sufficient for the observation of this effect is the use of LaF₃ as the substrate. This ionic solid has been proven to provide very efficient gating, while maintaining a good stability. They propose a model, which relies on the quantum capacitance of the doped monolayers, to explain the experimental observations.

The results are novel and interesting. They have the potential to provide a significant impact to the field and, more generally, could be appreciated also by the broad readership of Nature

Communications. The sample fabrication, experiments and data interpretation are conducted in a clear and consistent way. I would suggest to the authors to submit a revised version of the manuscript with some improvements and clarifications listed below to improve the quality of their work.

We thank the reviewer for the recognition of our work. We also greatly appreciate the reviewer's effort to help us clarify our points and improve on our paper.

1. The first question which “spontaneously” arises when reading the paper is whether the authors gave a fair share to the hypothesis that the low energy peak in their PL spectrum of the heterostructure is related to the recombination of the interlayer exciton. This quasiparticle complex has already been reported in the heterostructure studied by the authors in Nano Letters 17, 5342 and ACS Nano 9, 6459 among others. Because of the peculiar band alignment of the two constituents, the PL energy of the interlayer exciton in this heterostructure is only slightly lower in energy than the resonance of the intralayer MoSe₂ exciton, and the energy shift between the interlayer exciton and the MoSe₂ exciton reported in the literature is compatible with what is shown in figure 1c of the submitted manuscript. Additionally, the enhanced intensity of the low energy PL peak described in the manuscript could be explained in terms of a stronger absorption when the excitation is performed in resonance with intralayer excitons. The authors do not mention at all the interlayer exciton as the origin of the low energy peak in the PL spectrum. I find this quite surprising and I suggest instead that a complete discussion is presented, with convincing arguments which rule out the role of the interlayer excitons. A possibility could be for example to show temperature dependent data, similar to what is shown in Nature Physics 14, 801. This measurement could help clarify the origin of the peaks in the PL spectrum.

We much appreciate the reviewer's insightful question, and we take the opportunity here to elaborate on the assignment of the lower energy PL to the intralayer exciton in MoSe₂. The reviewer is correct that the interlayer exciton and intralayer exciton energy will be very close, and the PL position itself cannot be used as a reliable way to confirm one or the other.

As shown in our reply to the comment 1 of the reviewer 1, the TRPL clearly shows that the lifetime of the low energy PL peak from the heterojunction is similar to the intralayer exciton from MoSe₂, and both are significantly shorter than the reported dynamics of the interlayer exciton, ~ 80 fs at room temperature (Nanoscale, 2015, 7, 17523–17528).

In addition, we found that when both MoSe₂ and WS₂ layers are close to intrinsic, the PL from the lower energy is always quenched compared to the PL from monolayer MoSe₂. It is different from the reference ACS Nano 9, 6459, which shows an enhanced PL from the lower energy PL peak compared with the monolayer MoSe₂.

The reviewer's suggestion of low-temperature PL measurement, similar to what is done in Ref. Nature Physics 14, 801, is particularly insightful, since the interlayer exciton in the randomly stacked heterojunction, if exists at all, will be of indirect bandgap nature due to the K space misalignment and should show similar temperature behavior as reported in Nature Physics 14,

801. Due to the frozen ion effect in the LaF_3 , we cannot tune the doping easily through gate voltage at low temperature and did not do extensive low-temperature study. However, we did fix the gate voltage at 0 V and measure PL from the heterojunction at room temperature and 77 K under the same excitation condition. The results are shown in Fig. R6. It is evident that the PL at 77 K is much stronger than that at room temperature, opposite to the observation in Ref. Nature Physics 14, 801, and contradicts the expected behavior of indirect bandgap interlayer exciton.

With all the evidence, we confidently assign the low energy peak to the intralayer exciton in MoSe_2 . We attribute the absence of the interlayer exciton in the heterojunction to intentionally random stacking, which misaligns the two valleys of the two monolayers in the momentum space. We have added the above discussion to SI Supplementary Note 4. We thank the reviewer for helping us improve our manuscript.

Figure R6. PL spectra from the $\text{MoSe}_2/\text{WSe}_2$ junction on LaF_3 substrate at room temperature (red) and 77 K (blue).

2. Related to this, from lines 77-78, I seem to understand that all the measurements are performed at room temperature, although the authors never specify this explicitly. As per Nature journals template, they should add a “Method” section, where they indicate the experimental conditions they used and add more details concerning the device fabrication.

All the measurements in this paper are performed at room temperature. We have added a Method section about the detailed device fabrication and measurements. We thank the reviewer for the suggestion.

For the convenience of the reviewer, we also include the added Method section here:

Method

Device Fabrication

The MoSe₂/WS₂ heterostructure devices were fabricated through a layer-by-layer dry transfer technique²⁹. More specifically, each of the monolayer TMDC was sequentially transferred to the LaF₃ substrate, and a final BN flake was used to cap the heterostructure. Two pieces of few-layer graphene were used as the electrodes to contact the monolayer MoSe₂ and WS₂ layer separately, and both were grounded during the measurements, as schematically shown in Fig. 1b. The final devices were annealed in vacuum at 100 °C for 3 hours.

Optical Measurements

All the optical measurements in this work were performed at room temperature. The micro-PL measurements were performed with a home-built confocal microscope, in which the excitation lasers were focused to a spot size of ~ 2 μm. The PLE spectra were taken with a supercontinuum white laser (Fianium), and the filtered light (with bandwidth ~ 4 nm) was used as the excitation source."

3. Always related to my first question: did the authors measure PLE vs gate voltage also for excitation energies >2eV? Does the intensity of the low energy peak stay constant or does it decrease?

We have measured the PL for excitation energy > 2 eV, as shown in Fig. 3a,b in the main text. For example, at the gate voltage of -2 V (Fig. 3a in the main text), the PL intensity of the lower energy peak in the heterojunction (red) reaches a local maximum at excitation energy ~ 2 eV (the WS₂ A exciton resonance) and decreases as the excitation energy further increases. A similar trend is observed at the gate voltage of 4 V (Fig. 3b, red curve) with even more pronounced change. These behaviors can be naturally understood from the charge transfer between the two layers: Once WS₂ is optically excited, the hole transfer from WS₂ to MoSe₂ will enhance the PL from MoSe₂ in the heterojunction. Such enhancement is proportional to the optical absorption of WS₂ at the given excitation energy. As a result, the PLE spectrum (i.e., excitation energy-dependent PL intensity) will locally follow the absorption spectrum of WS₂ (plus a background PL intensity from MoSe₂ absorption itself) and shows a local maximum at the WS₂ A exciton resonance of ~2 eV.

4. It would be very instructive if the authors could quantify the doping level reached by using the LaF₃ substrate and compare them with SiO₂-based devices.

It has been demonstrated previously [Nano Lett. 2018, 18, 2387-2392] that the LaF₃ substrate can gate few-layer MoS₂ devices with an effective capacitance of $\sim 4 \mu\text{F}/\text{cm}^2$, which is ~ 400 times more efficient than the 300 nm SiO₂ device. In our devices shown in this work, we cannot directly measure the capacitance. However, we have fabricated a monolayer MoS₂ FET device gated by LaF₃, which shows on-state conductance similar to the previous work, shown in Fig. R7.

Figure R7. Two-terminal monolayer MoS₂ device gated by LaF₃ back gate. (a) Conductance as a function of the gate voltage. (b) Conductance as a function of temperature showing an insulator (-3 V, -2 V) to metal (-1 V, 0 V) transition when the gate voltage increase.

We fit the conductance (G) as a function of the gate voltage (V_g) with the following expression:

$$G = \frac{1}{\frac{L}{WC\mu(V_g - V_T)} + R_c},$$

where C is the capacitance, R_c is the contact resistance, V_T is the turning-on voltage, μ is the electron mobility, L and W are the length and the width of the channel, respectively.

By fitting the G-V_g curve with this model (black dotted line in Fig. R7), we obtained $C = 1.41 \mu\text{F}/\text{cm}^2$ and $n = 2.36 \times 10^{13} \text{ cm}^{-2}$ at $V_g = 0 \text{ V}$. By comparison, Wu et al.'s results [Nano Lett. 2018, 18, 2387-2392] show $C = 4 \mu\text{F}/\text{cm}^2$ and $n \sim 1 \times 10^{13} \text{ cm}^{-2}$ at $V_g = 3 \text{ V}$. We thus believe that the gating efficiency of our LaF₃ substrate should be similar to the previous report and is rough ~ 140 times more efficient than the 300 nm SiO₂/Si substrate.

We have also revised the main text accordingly as follows:

“This observation suggests that the efficient gating from LaF₃ is essential for realizing the PL enhancement in the heterojunction. Previous work has shown that the LaF₃ back gate should be at least more than 100 times more efficient than the silicon back gate with 300 nm thermal oxide³⁰.”

5. I was wandering what role can play the charged exciton in the absorption of WS₂. If the gating is so efficient, at a sufficient gate voltage there should be a “transfer” of oscillator strength from the neutral to the charged exciton, which could be for example visualized by performing gate

dependent reflectivity contrast map (as in PRB 96, 085302, for example), which should in turn influence the energy at which the resonance is detected in PLE.

Indeed charged exciton (trion) will emerge when WS_2 is sufficiently n-doped. However, in the current model, the charged exciton (trion) plays the same role as the neutral exciton. For example, whether exciting an exciton or a trion in WS_2 would lead to the same amount of additional electron and hole density, and therefore the same amount of charge transfer between layers. As a result, the only relevant parameter here is the absorption of WS_2 at the excitation light energy, regardless of the nature of the excited state. The introduction of charged exciton does not change our theoretical understanding of our model.

Indeed, the trion formation can lead to a shift of WS_2 optical resonance. We show the reflectance spectra for different gate voltages in Fig. R8. Because the exciton linewidth here (~ 200 meV) is much broader than the typical exciton-trion separation in WS_2 (~ 30 meV [Nature Communications 4, 1474 (2013)]), we cannot separate the exciton and trion resonance, and the doping-induced resonance shift is also negligible.

Figure R8. Reflectance spectra for the gate voltage of -3 V (a) and 4 V (b).

6. Given that there are no space restrictions, I think that the readability of the manuscript would be improved if the authors provided more detail about the derivation of the equivalent circuit shown as the inset of figure 4.

We thank the reviewer for the suggestion and take the opportunity to elaborate on the derivation of the equivalent circuit shown in the inset of Fig. 4 in the main text.

It has been shown by Luryi (APL 52, 501 (1988)) that for a circuit shown in Fig. R9a, the equivalent capacitance circuit is as shown in Fig. R9b. In Fig. R9a, the top and bottom plates are ideal metal plate, and the middle plate, Q, is a two-dimensional metal which does not screen the field completely. It could be 2D electron gas (2DEG) of a quantum well (QW), or in our case, a monolayer TMDC. As the quantum capacitance, $C_Q = g \frac{me^2}{\pi h^2}$, is comparable to geometry capacitance C_1 and C_2 , the energy to fill electron in the TMDC is not negligible and has to be

taken into account. By minimizing the total energy of the system, including the energy to charge the two geometry capacitors (C_1 and C_2) and quantum capacitor (C_Q), it can be shown that the charge density in electrode 1 (σ_1) and electrode 2 (σ_2) follows the relation, $\sigma_2 = -\sigma_1 \frac{C_2}{C_2+C_Q}$, and the charge neutrality condition gives $\sigma_Q = -\sigma_1 - \sigma_2$. It is evident then that the equivalent circuit of Fig. R9a should be Fig. R9b.

Figure R9. Schematic of the capacitance model for a two-dimensional material sandwiched by two plates (a) and the equivalent circuit (b) [APL 52, 501 (1988)].

Now, if we only have one monolayer TMDC device shown as schematically in Fig. R10a, it will be equivalent to have the bottom plate (the grounded C_2) in Fig. R9a to be placed at infinity, which gives a $C_2 = 0$, and the effective capacitance will be C_1 and C_Q in series. For the heterostructure device schematically shown in Fig. 10c (also Fig. 1b of the main text), the bottom plate is not an ideal metal anymore and does not completely screen the electrical field. Instead, it is a quantum plate, just like the middle plate Q. In this case, we name the middle plate Q_1 , and following our equivalent circuit of the monolayer device (Fig. 10b), now the C_2 in Fig. R9b should be replaced with geometry capacitance (C_2) and quantum capacitance (C_{Q2}) in series, and the resulting equivalent circuit will be as shown in Fig. R10d (also the inset of Fig. 4 of the main text). Another way to understand is that, the extra voltage drop on the quantum capacitance 1 (C_{Q1}) has to be the same as the total voltage drop on the geometry capacitance (C_2) and quantum capacitance (C_{Q2}), as both the quantum plate 1 (the original middle plate Q in Fig. 10a) and quantum plate 2 (the original bottom plate in Fig. 10a) are both grounded, same as our experimental setup.

Figure R10. (a) and (b) are the schematic for the monolayer MoSe₂ device and its corresponding equivalent capacitance circuit. (c) and (d) are the schematic for the heterostructure device and its corresponding equivalent capacitance circuit.

We have included in the derivation in SI Supplementary Note 3. In the main text, we refer the readers to the derivation in the SI as follows:

“Taking into account the quantum capacitance of the monolayer MoSe₂ and WS₂, for the device configuration shown in Fig. 1b, the effective capacitance model can be schematically shown as the inset of Fig. 4 (MoSe₂ being the bottom layer, and detailed derivation in SI Supplementary Note 3).”

7. The authors wrote they fabricated seven devices with LaF₃ as the substrate. Did they perform similar measurements as a check on any other device? Do they reproduce their observations?

We thank the reviewer for the suggestion. The PL EF with no gate voltage applied for seven different devices has been shown in our reply to comment 1 of reviewer 1 (Fig. R2). Here we also provide a PLE study of a second device on LaF₃ in Fig. R10, which shows similar results as Fig. 3 in the main text. We also include this information in the revised SI as Supplementary Note 6.

Figure R11. PLE spectra and EF as a function of the excitation for a second device on LaF_3 . (a)-(c) are the PL spectra for the gate voltage of -3 V, 0 V, and 4 V. The corresponding EF as a function of the excitation photon energy is shown in (d)-(f).

8. Small points: line 91. The authors write WSe_2 , I suppose it should be WS_2 . Lines 95-96. The authors write about an “evident” enhancement of the signal of MoSe_2 . I think it is misleading for the reader: one might think that the enhancement occurs for the signal from the heterostructure when the voltage is changed, which is not the case. The authors refer to the larger intensity of the signal from the heterostructure as compared to the signal from MoSe_2 at the same gate voltage. I suggest to rephrase to avoid any confusion.

We sincerely thank the reviewer for spotting the typos. We have corrected them in the main text (highlighted).

We agree with the reviewer that our original writing about the “evident” enhancement is confusing, and we have revised it as following in the main text:

“We can see from the color plots (Fig. 2a, b) that, although the MoSe_2 A exciton PL intensity is weaker in $\text{MoSe}_2/\text{WS}_2$ heterojunction (Fig. 2b) than in the monolayer MoSe_2 (Fig. 2a) for the gate voltage from ~ -2 V to -1 V, the PL is stronger in the heterojunction than in the monolayer MoSe_2 at the gate voltage > 0 V. This relative PL ratio from quenching to enhancement transition is clearly illustrated in the PL spectra in Fig. 2d, which combine the line cuts of Fig. 2a and Fig. 2b at the gate voltage -2 V and 4 V.”

We thank the reviewer for helping us to improve our manuscript.

REVIEWERS' COMMENTS:

Reviewer #1 (Remarks to the Author):

I am reviewer 1. The authors have addressed all my concerns and I would recommend the publication of this manuscript.

Reviewer #2 (Remarks to the Author):

The authors have substantiated their claims and clarified their points responding satisfactorily to the reviewers' questions and doubts. I believe the current version of the main manuscript and (especially) the supplementary material are significantly improved as compared to the first submitted version.

I can therefore recommend this manuscript for publication in Nature Communications.

Dear Reviewers,

We are glad that both reviewers are satisfied with our revision of manuscript and the supplementary information. We greatly appreciate both reviewers' efforts and comments that helped us to improve our manuscript.

Reviewer #1 (Remarks to the Author):

I am reviewer 1. The authors have addressed all my concerns and I would recommend the publication of this manuscript.

Reviewer #2 (Remarks to the Author):

The authors have substantiated their claims and clarified their points responding satisfactorily to the reviewers' questions and doubts. I believe the current version of the main manuscript and (especially) the supplementary material are significantly improved as compared to the first submitted version.

I can therefore recommend this manuscript for publication in Nature Communications.